# Fair Numerical Algorithm of Coset Cardinality Spectrum for Distributed Arithmetic Coding

**DOI:** 10.3390/e25030437

**Published:** 2023-03-01

**Authors:** Yong Fang, Nan Yang

**Affiliations:** School of Information Engineering, Chang’an University, Xi’an 710064, China

**Keywords:** distributed arithmetic coding, Slepian-Wolf coding, coset cardinality spectrum, numerical algorithm

## Abstract

As a typical symbol-wise solution of asymmetric Slepian-Wolf coding problem, Distributed Arithmetic Coding (DAC) non-linearly partitions source space into disjoint cosets with unequal sizes. The distribution of DAC coset cardinalities, named the Coset Cardinality Spectrum (CCS), plays an important role in both theoretical understanding and decoder design for DAC. In general, CCS cannot be calculated directly. Instead, a numerical algorithm is usually used to obtain an approximation. This paper first finds that the contemporary numerical algorithm of CCS is theoretically imperfect and does not finally converge to the real CCS. Further, to solve this problem, we refine the original numerical algorithm based on rigorous theoretical analyses. Experimental results verify that the refined numerical algorithm amends the drawbacks of the original version.

## 1. Introduction

As an important branch of network information theory, *Distributed Source Coding* (DSC) can find broad potential applications in many scenarios (e.g., wireless sensor network, genome compression, etc.). Just as traditional source coding, DSC has two forms. Lossless DSC is also called *Slepian-Wolf Coding* (SWC) [1]. The general form of lossy DSC is referred to as Berger-Tung coding [2], while a special case of asymmetric lossy DSC with side information at the decoder is referred to as Wyner-Ziv Coding (WZC) [3]. Up to now, the most important works on SWC have focused on its asymmetric form. Let *X* and *Y* be two correlated discrete random variables. Given *Y* available only at the decoder, the achievable rates of compressing *X* without loss are bounded by R≥H(X|Y). Different from SWC, an important property of WZC is that it usually suffers rate loss when compared to traditional lossy coding with side information available at both the encoder and the decoder. However, if the difference between source and side information is an independent Gaussian random variable, there is no rate loss [4,5].

This paper treats only asymmetric SWC. The asymmetric SWC problem is in essence a channel coding problem [6,7,8,9]. To show this point, one can take *Y* as a noisy version of *X* corrupted by virtual channel noise and take the bitstream of *X* as the index of the coset containing *X*. If the elements in each coset are spaced as far (in Hamming distance) as possible and *Y* is near (in Hamming distance) enough to *X*, then *X* can be recovered from its bitstream with the help of *Y*. From this viewpoint, the bitstream of *X* is actually the syndrome of a coset code. Hence traditionally, asymmetric SWC was implemented by channel codes (e.g., Turbo codes [10], *Low-Density Parity-Check* (LDPC) codes [11], and polar codes [12], etc.).

Arithmetic coding is usually deemed as the most important method for lossless data compression [13,14]. Due to the duality between source coding and channel coding, arithmetic coding can be easily modified to achieve the purpose of error detection and correction [15,16,17]. In 2007, people found that after some slight modifications, arithmetic coding can also serve as a coset code to implement asymmetric SWC. There are mainly two approaches: One is *bit puncturing* [18] and the other is *interval enlarging* [19,20,21]. This paper will focus only on the latter approach, which is often referred to as *Distributed Arithmetic Coding* (DAC), while ignoring the former. For *independent and identically-distributed* (i.i.d.) binary sources, DAC is in general inferior to channel codes, e.g., LDPC codes, polar codes, etc., as shown by the experimental results in [22]. However, for binary sources with memory or nonbinary sources, DAC performs significantly better than channel codes [23,24,25,26].

In nature, DAC is a many-to-one nonlinear mapping that partitions source space into disjoint cosets of unequal cardinalities. Then an important problem is how DAC coset cardinality is distributed, which can be answered by the so-called *Coset Cardinality Spectrum* (CCS). The idea of CCS budded in [27,28] and was formally defined for uniform and nonuniform binary sources in [29,30], respectively. The concept of CCS is very useful as it not only can serve as a theoretical tool to analyze the properties of DAC [31,32], but also can be used to derive correct decoding formulae [31,32] (See the discussion in the last paragraph of Section 2).

If the stream of source symbols is grouped into length-*n* blocks, then DAC CCS is a tuple of n+1
*probability density functions* (pdfs). It is impossible to deduce the exact closed form for each pdf directly. Instead, one can begin with the final pdf, which is usually simple and calculable, and then derive each pdf via a backward recursion. In [29], a numerical algorithm was proposed to implement the backward recursion for uniform binary sources, and then it was generalized to nonuniform binary sources in [30].

However, the numerical algorithm proposed in [29,30] is very primitive and lacks theoretical justification. This paper will make an in-depth analysis of the numerical algorithm proposed in [29,30] and observe the results carefully. It will be found that the numerical algorithm proposed in [29,30] does not exactly converge to the real CCS. After a strict theoretical analysis, this paper will propose a novel numerical algorithm, which perfectly overcomes the drawbacks of the original numerical algorithm.

The rest of this paper is arranged as below. Section 2 briefly reviews the background knowledge of DAC and CCS. Section 3 presents the original numerical algorithm and its trivial upgrade version. Section 4 proposes the novel numerical algorithm for DAC CCS based on solid theoretical analyses. Section 5 reports some experimental results to compare three numerical algorithms. Finally, Section 6 concludes this paper.

## 2. Review on DAC and CCS

Let $X^{n}≜\left(X_{1},\ldots ,X_{n}\right)$ be a length-*n* binary source block with bias probability $\mathrm{Pr}(X_{i}=1)=p$. The entropy of *X* is $H\left(X\right)=-p\log_{2}p-\left(1-p\right)\log_{2}\left(1-p\right)$. The DAC codec recursively maps every source symbol onto an interval in [0,1) according to the following rule
(1)$$\begin{array}{c} x\in B≜\left\{0,1\right\}\to \left[x\left(1-p^{r}\right),\left(1-x\right){\left(1-p\right)}^{r}+x\right)\subset \left[0,1\right), \end{array}$$
where r∈[0,1] is called the normalized rate or overlapping factor. Let $\left[l\left(X^{n}\right),h\left(X^{n}\right)\right)\subset \left[0,1\right)$ denote the mapping interval of $X^{n}$. In theory, $\left[l\left(X^{n}\right),h\left(X^{n}\right)\right)$ can be represented by $-\log_{2}\left(h\left(X^{n}\right)-l\left(X^{n}\right)\right)\in R$ bits. However, due to the indivisibility of a bit, the bitstream of $X^{n}$ actually includes $-\lfloor \log_{2}\left(h\left(X^{n}\right)-l\left(X^{n}\right)\right)\rfloor \in Z$ bits, which can be explained as a real number $U_{0}$ in $\left[l\left(X^{n}\right),h\left(X^{n}\right)\right)$. Hence, there is a rate loss of δ bits, where
(2)$$\begin{array}{c} \delta ≜\log_{2}\left(h\left(X^{n}\right)-l\left(X^{n}\right)\right)-\left\lfloor \log_{2}\left(h\left(X^{n}\right)-l\left(X^{n}\right)\right)\right\rfloor \in \left[0,1\right). \end{array}$$If we decode the bitstream $U_{0}$ along the path $X^{n}$, then we will obtain a tuple of n+1 real numbers $U_{0}^{n}≜\left(U_{0},U_{1},\cdots ,U_{n}\right)$ which can be deduced recursively. The forward recursion of $U_{0}^{n}$ can be easily transformed into an equivalent backward recursion as shown below [30].
(3)$$\begin{array}{c} \left\{\begin{array}{cc} U_{i-1}={\left(1-p\right)}^{r}U_{i}, & X_{i}=0\\ U_{i-1}=p^{r}U_{i}+\left(1-p^{r}\right), & X_{i}=1 \end{array}\right.. \end{array}$$The pdf of $U_{i}$ is called the *i*-th CCS and denoted by $f_{n,i}\left(u\right)$, 0≤u<1, which is usually simplified as $f_{i}\left(u\right)$. Especially, $f_{0}\left(u\right)$ is called the initial CCS and $f_{n}\left(u\right)$ is called the final CCS.

At this point, we have two choices: One is beginning from the initial CCS $f_{0}\left(u\right)$ and then deducing $f_{i}\left(u\right)$ for 0<i≤n via a forward recursion; the other is beginning from the final CCS $f_{n}\left(u\right)$ and then deducing $f_{i}\left(u\right)$ for 0≤i<n via a backward recursion. Unfortunately, it is impossible to deduce the initial CCS $f_{0}\left(u\right)$ directly, so forward recursion is infeasible. Instead, it is proved in [30] that depending on the parameters *p* and *r*, $f_{n}\left(u\right)$ will tend to be a piecewise uniform function or
(4)$$\begin{array}{c} \lim_{n\to \infty }f_{n}\left(u\right)=\frac{\log_{2}e}{1+|1-2u|}. \end{array}$$Once $f_{n}\left(u\right)$ is known, $f_{i}\left(u\right)$ for 0≤i<n can be deduced via a backward recursion [30].
(5)$$\begin{array}{c} f_{i-1}\left(u\right)={\left(1-p\right)}^{1-r}f_{i}\left(u{\left(1-p\right)}^{-r}\right)+p^{1-r}f_{i}\left(\left(u-\left(1-p^{r}\right)\right)p^{-r}\right). \end{array}$$In general, it is impossible to deduce $f_{i-1}\left(u\right)$ from $f_{i}\left(u\right)$ analytically, even though the closed form of $f_{i}\left(u\right)$ is known. Hence, we have to resort to a numerical algorithm, which mimics (Equation 5) in a discrete way.

Regarding the physical meaning of CCS, there is a detailed explanation in [28]. Let us take uniform binary sources as an example. For any real number u∈[0,1), if *u* is fed into a DAC decoder with overlapping factor *r*, then we will get an incomplete binary tree. In this tree, the number of level-*i* nodes is roughly $\left(\lim_{n\to \infty }f_{n,n-i}\right)\cdot 2^{i(1-r)}$. If we implement the decoder in a breadth-first way, i.e., with the *M*-algorithm, for every level-*i* node, the sub-tree grown from this level-*i* node will have about $\left(\lim_{n\to \infty }f_{n,i}\left(u\right)\right)\cdot 2^{\left(n-i)(1-r\right)}$ leaf nodes, where *u* is the real number at the level-*i* node. Obviously, those level-*i* nodes with larger $\left(\lim_{n\to \infty }f_{n,i}\left(u\right)\right)$ should be more likely, and vice versa. That is why DAC CCS can be used to derive correct decoding formulae.

## 3. Original Numerical Algorithms

### 3.1. Rounding Numerical Algorithm

The first version of the numerical algorithm was proposed in [28,30] for uniform and nonuniform binary sources, respectively. This algorithm divides the interval [0,1) into *N* segments and then uses a finite number of $f_{i}\left(j/N\right)$’s, where j∈[0:N)≜{0,…,N−1}, to approximate $f_{i}\left(u\right)$. For simplicity, we use ${\hat{f}}_{i}\left(j\right)$ to denote the approximation of $f_{i}\left(j/N\right)$. Then (Equation 5) can be discretized as
(6)$$\begin{array}{c} {\hat{f}}_{i-1}\left(j\right)={\left(1-p\right)}^{1-r}{\hat{f}}_{i}\left(j^{\prime }\right)+p^{1-r}{\hat{f}}_{i}\left(j^{\prime \prime }\right). \end{array}$$Now the key is to find a good mapping from *j* to $j^{\prime }$ and $j^{\prime \prime }$. Let ⌊·⌉ denote the rounding operation. This problem was solved by a brute-force method in [28,30] as follows:(7)$$\begin{array}{c} \left\{\begin{array}{c} j^{\prime }=\left\lfloor j{\left(1-p\right)}^{-r}\right\rceil \\ j^{\prime \prime }=\left\lfloor \left(j-N\left(1-p^{r}\right)\right)p^{-r}\right\rceil \end{array}\right.. \end{array}$$For some j∈[0:N), we will have $j^{\prime }\notin \left[0:N\right)$ or $j^{\prime \prime }\notin \left[0:N\right)$. It does not matter because we have $\hat{f}\left(j\right)\equiv 0$ for any j∉[0:N) according to the definition of CCS. Since the core of (Equation 7) is the rounding operation, this method will be formally referred to as *rounding numerical algorithm* below.

Though the rounding numerical algorithm works well at first glance [28,30], its rationality is indeed flawed. On one hand, since ${\left(1-p\right)}^{-r}>1$, we have $j{\left(1-p\right)}^{-r}\geq j$ and thus among *N* points of ${\hat{f}}_{i}\left(j\right)$’s, only about $N{\left(1-p\right)}^{r}<N$ points, i.e., $0\leq j<j_{\max}\approx N{\left(1-p\right)}^{r}$, are used for the first line of (Equation 7). On the other hand,
(8)$$\begin{array}{cc} \left(j-N\left(1-p^{r}\right)\right)p^{-r}\; & =N-\left(N-j\right)p^{-r}\\ \; & =\left(N-j\right)-\left(N-j\right)p^{-r}+j\\ \; & =\left(N-j\right)\left(1-p^{-r}\right)+j<j, \end{array}$$
where the last inequality is due to N>j and $p^{-r}>1$. Thus among *N* points of ${\hat{f}}_{i}\left(j\right)$’s, only about $Np^{r}<N$ points, i.e., $N\left(1-p^{r}\right)\approx j_{\min}\leq j<N$, are used for the second line of (Equation 7). In summary, not all *N* points of ${\hat{f}}_{i}\left(j\right)$’s will be used to generate ${\hat{f}}_{i-1}\left(j\right)$’s. In other words, some points of ${\hat{f}}_{i}\left(j\right)$’s are discarded without being used when we try to deduce ${\hat{f}}_{i-1}\left(j\right)$ according to (Equation 6), which is equivalent to the case that partial information of $f_{i}\left(u\right)$ is lost when we try to deduce $f_{i-1}\left(u\right)$ according to (Equation 5). This is imperfect in theory and may bring two negative effects in practice:The rounding numerical algorithm cannot generate an accurate approximation of CCS. This phenomenon will be observed in the experimental results of Section 5.Even if $\sum_{j=0}^{N-1}{\hat{f}}_{i}\left(j\right)=N$, (Equation 6) does not strictly satisfy the normalization condition $\sum_{j=0}^{N-1}{\hat{f}}_{i-1}\left(j\right)=N$. Thus an extra re-normalization step is needed after (Equation 6).

### 3.2. Linear Numerical Algorithm

To improve the performance of the rounding numerical algorithm, more points of ${\hat{f}}_{i}\left(j\right)$’s should be involved in generating ${\hat{f}}_{i-1}\left(j\right)$’s. This problem can be solved by linear interpolation. Below we will propose a trivial upgrade of the rounding numerical algorithm, which we will refer to as *linear numerical algorithm*. Let ⌊·⌋ and ⌈·⌉ denote the flooring and ceiling operations, respectively. Let us define $\alpha ≜j{\left(1-p\right)}^{-r}\geq j$ and $\beta ≜\left(j-N\left(1-p^{r}\right)\right)p^{-r}<j$. In general, (Equation 6) can be refined as
(9)$$\begin{array}{c} {\hat{f}}_{i-1}\left(j\right)={\left(1-p\right)}^{1-r}g_{0}\left(\alpha \right)+p^{1-r}g_{1}\left(\beta \right), \end{array}$$
where
(10)$$\begin{array}{c} \left\{\begin{array}{c} g_{0}\left(\alpha \right)=\left(\left\lceil \alpha \right\rceil -\alpha \right){\hat{f}}_{i}\left(\left\lfloor \alpha \right\rfloor \right)+\left(\alpha -\left\lfloor \alpha \right\rfloor \right){\hat{f}}_{i}\left(\left\lceil \alpha \right\rceil \right)\\ g_{1}\left(\beta \right)=\left(\left\lceil \beta \right\rceil -\beta \right){\hat{f}}_{i}\left(\left\lfloor \beta \right\rfloor \right)+\left(\beta -\left\lfloor \beta \right\rfloor \right){\hat{f}}_{i}\left(\left\lceil \beta \right\rceil \right) \end{array}\right.. \end{array}$$Then we discuss some special cases:If α∈Z, then $g_{0}\left(\alpha \right)={\hat{f}}_{i}\left(\alpha \right)$. If β∈Z, then $g_{1}\left(\beta \right)={\hat{f}}_{i}\left(\beta \right)$.If ⌊α⌋≥N, then $g_{0}\left(\alpha \right)=0$. If ⌈β⌉<0, then $g_{1}\left(\beta \right)=0$.If ⌊α⌋<N and ⌈α⌉≥N, then $g_{0}\left(\alpha \right)={\hat{f}}_{i}\left(\left\lfloor \alpha \right\rfloor \right)$. If ⌊β⌋<0 and ⌈β⌉≥0, then $g_{1}\left(\beta \right)={\hat{f}}_{i}\left(\left\lceil \beta \right\rceil \right)$.

However, it must be pointed out that the linear numerical algorithm still does not guarantee that all *N* points of ${\hat{f}}_{i}\left(j\right)$’s are used to generate ${\hat{f}}_{i-1}\left(j\right)$’s, and a renormalization step is still needed after (Equation 9).

## 4. Fair Numerical Algorithm

In the rounding/linear numerical algorithms, we take ${\hat{f}}_{i}\left(j\right)$ as an approximation of $f_{i}\left(j/N\right)$, i.e., ${\hat{f}}_{i}\left(j\right)\approx f_{i}\left(j/N\right)$ for any j∈[0:N), and for each ${\hat{f}}_{i-1}\left(j\right)$, we try to find the corresponding ${\hat{f}}_{i}\left(j^{\prime }\right)$. This understanding is however incorrect. The correct physical meaning of ${\hat{f}}_{i}\left(j\right)$ should be an approximation of the scaled-up probability of $U_{i}$ falling into the interval [j/N,(j+1)/N)), i.e.,
(11)$$\begin{array}{c} {\hat{f}}_{i}\left(j\right)\approx N\int_{j/N}^{(j+1)/N}f_{i}\left(u\right)\;du. \end{array}$$According to (Equation 5), we have
(12)$$\begin{array}{c} \int_{j/N}^{(j+1)/N}f_{i-1}\left(u\right)\;du=\int_{j/N}^{(j+1)/N}\left({\left(1-p\right)}^{1-r}f_{i}\left(u{\left(1-p\right)}^{-r}\right)+p^{1-r}f_{i}\left(\left(u-\left(1-p^{r}\right)\right)p^{-r}\right)\right)\;du. \end{array}$$Let $u^{\prime }≜u{\left(1-p\right)}^{-r}$ and $u^{\prime \prime }≜\left(u-\left(1-p^{r}\right)\right)p^{-r}$. Let
(13)$$\begin{array}{c} \left\{\begin{array}{c} I_{0}≜\left[j{\left(1-p\right)}^{-r}/N,\left(j+1\right){\left(1-p\right)}^{-r}/N\right)\\ I_{1}≜\left[\left(j-N\left(1-p^{r}\right)\right)p^{-r}/N,\left(j+1-N\left(1-p^{r}\right)\right)p^{-r}/N\right) \end{array}\right.. \end{array}$$Then we can obtain
(14)$$\begin{array}{cc} \int_{j/N}^{(j+1)/N}f_{i-1}\left(u\right)\;du & =\left(1-p\right)\int_{u^{\prime }\in I_{0}}f_{i}\left(u^{\prime }\right)\;du^{\prime }+p\int_{u^{\prime \prime }\in I_{1}}f_{i}\left(u^{\prime \prime }\right)\;du^{\prime \prime }\\ \; & =\left(1-p\right)\int_{u\in I_{0}}f_{i}\left(u\right)\;du+p\int_{u\in I_{1}}f_{i}\left(u\right)\;du. \end{array}$$Naturally, we have
(15)$$\begin{array}{c} {\hat{f}}_{i-1}\left(j\right)=\left(1-p\right)\cdot g_{0}\left(j\right)+p\cdot g_{1}\left(j\right), \end{array}$$
where
(16)$$\begin{array}{c} \left\{\begin{array}{c} g_{0}\left(j\right)\approx N\int_{u\in I_{0}}f_{i}\left(u\right)\;du\\ g_{1}\left(j\right)\approx N\int_{u\in I_{1}}f_{i}\left(u\right)\;du \end{array}\right.. \end{array}$$In plain words, $g_{0}\left(j\right)$ is an approximation of the scaled-up probability of $U_{i}$ falling into the interval $I_{0}$, and $g_{1}\left(j\right)$ is an approximation of the scaled-up probability of $U_{i}$ falling into the interval $I_{1}$.

### 4.1. Calculation of $g_{0}\left(j\right)$

Let $\lambda_{0}≜j{\left(1-p\right)}^{-r}\geq 0$ and $\eta_{0}≜\left(j+1\right){\left(1-p\right)}^{-r}>\lambda_{0}\geq 0$. Then we can obtain
(17)$$\begin{array}{c} N\int_{u\in I_{0}}f_{i}\left(u\right)\;du=N\int_{\lambda_{0}/N}^{(\left\lfloor \lambda_{0}\right\rfloor +1)/N}f_{i}\left(u\right)\;du+N\int_{(\left\lfloor \lambda_{0}\right\rfloor +1)/N}^{(\left\lfloor \lambda_{0}\right\rfloor +2)/N}f_{i}\left(u\right)\;du+\cdots \\ +N\int_{(\left\lfloor \eta_{0}\right\rfloor -1)/N}^{\lfloor \eta_{0}\rfloor /N}f_{i}\left(u\right)\;du+N\int_{\lfloor \eta_{0}\rfloor /N}^{\eta_{0}/N}f_{i}\left(u\right)\;du. \end{array}$$For *N* sufficiently large, $f_{i}\left(u\right)$ can be taken as uniform over [j/N,(j+1)/N). Hence we have
(18)$$\begin{array}{cc} N\int_{\lambda_{0}/N}^{(\left\lfloor \lambda_{0}\right\rfloor +1)/N}f_{i}\left(u\right)\;du\; & \approx \left(\left\lfloor \lambda_{0}\right\rfloor +1-\lambda_{0}\right)\cdot N\int_{\lfloor \lambda_{0}\rfloor /N}^{(\left\lfloor \lambda_{0}\right\rfloor +1)/N}f_{i}\left(u\right)\;du\\ \; & \approx \left(\left\lfloor \lambda_{0}\right\rfloor +1-\lambda_{0}\right)\cdot {\hat{f}}_{i}\left(\left\lfloor \lambda_{0}\right\rfloor \right) \end{array}$$
and
(19)$$\begin{array}{cc} N\int_{\lfloor \eta_{0}\rfloor /N}^{\eta_{0}/N}f_{i}\left(u\right)\;du\; & \approx \left(\eta_{0}-\left\lfloor \eta_{0}\right\rfloor \right)\cdot N\int_{\lfloor \eta_{0}\rfloor /N}^{(\left\lfloor \eta_{0}\right\rfloor +1)/N}f_{i}\left(u\right)\;du\\ \; & \approx \left(\eta_{0}-\left\lfloor \eta_{0}\right\rfloor \right)\cdot {\hat{f}}_{i}\left(\left\lfloor \eta_{0}\right\rfloor \right). \end{array}$$According to the above analysis, we can obtain the following results.

$N\leq \lambda_{0}<\eta_{0}$: It is easy to know $g_{0}\left(j\right)\equiv 0$.$\lambda_{0}<N\leq \eta_{0}$: In general, we have
(20)$$\begin{array}{c} g_{0}\left(j\right)=\left(1-\left(\lambda_{0}-\left\lfloor \lambda_{0}\right\rfloor \right)\right)\cdot {\hat{f}}_{i}\left(\left\lfloor \lambda_{0}\right\rfloor \right)+\sum_{j^{\prime }=\left\lfloor \lambda_{0}\right\rfloor +1}^{N-1}{\hat{f}}_{i}\left(j^{\prime }\right). \end{array}$$Especially, if $\lambda_{0}\in Z$, then
(21)$$\begin{array}{c} g_{0}\left(j\right)=\sum_{j^{\prime }=\lambda_{0}}^{N-1}{\hat{f}}_{i}\left(j^{\prime }\right). \end{array}$$$\lambda_{0}<\eta_{0}<N$: In general, we have
(22)$$\begin{array}{c} g_{0}\left(j\right)=\left(1-\left(\lambda_{0}-\left\lfloor \lambda_{0}\right\rfloor \right)\right)\cdot {\hat{f}}_{i}\left(\left\lfloor \lambda_{0}\right\rfloor \right)+\left(\eta_{0}-\left\lfloor \eta_{0}\right\rfloor \right)\cdot {\hat{f}}_{i}\left(\left\lfloor \eta_{0}\right\rfloor \right)+\sum_{j^{\prime }=\left\lfloor \lambda_{0}\right\rfloor +1}^{\lfloor \eta_{0}\rfloor -1}{\hat{f}}_{i}\left(j^{\prime }\right). \end{array}$$Let us consider three special cases:—If $\lambda_{0}\in Z$ and $\eta_{0}\notin Z$, then
(23)$$\begin{array}{c} g_{0}\left(j\right)=\left(\eta_{0}-\left\lfloor \eta_{0}\right\rfloor \right)\cdot {\hat{f}}_{i}\left(\left\lfloor \eta_{0}\right\rfloor \right)+\sum_{j^{\prime }=\lambda_{0}}^{\lfloor \eta_{0}\rfloor -1}{\hat{f}}_{i}\left(j^{\prime }\right). \end{array}$$—If $\lambda_{0}\notin Z$ and $\eta_{0}\in Z$, then
(24)$$\begin{array}{c} g_{0}\left(j\right)=\left(\left\lceil \lambda_{0}\right\rceil -\lambda_{0}\right)\cdot {\hat{f}}_{i}\left(\left\lfloor \lambda_{0}\right\rfloor \right)+\sum_{j^{\prime }=\left\lceil \lambda_{0}\right\rceil }^{\eta_{0}-1}{\hat{f}}_{i}\left(j^{\prime }\right). \end{array}$$—If $\lambda_{0}\in Z$ and $\eta_{0}\in Z$, then
(25)$$\begin{array}{c} g_{0}\left(j\right)=\sum_{j^{\prime }=\lambda_{0}}^{\eta_{0}-1}{\hat{f}}_{i}\left(j^{\prime }\right). \end{array}$$

### 4.2. Calculation of $g_{1}\left(j\right)$

Let $\eta_{1}≜\left(j+1-N\left(1-p^{r}\right)\right)p^{-r}\leq N$ and $\lambda_{1}≜\left(j-N\left(1-p^{r}\right)\right)p^{-r}<\eta_{1}\leq N$. Similarly, we can get the following results.

$\lambda_{1}<\eta_{1}<0$: It is easy to know $g_{1}\left(j\right)\equiv 0$.$\lambda_{1}<0\leq \eta_{1}$: In general, we have
(26)$$\begin{array}{c} g_{1}\left(j\right)=\left(\eta_{1}-\left\lfloor \eta_{1}\right\rfloor \right)\cdot {\hat{f}}_{i}\left(\left\lfloor \eta_{1}\right\rfloor \right)+\sum_{j^{\prime }=0}^{\lfloor \eta_{1}\rfloor -1}{\hat{f}}_{i}\left(j^{\prime }\right). \end{array}$$Especially, if $\eta_{1}\in Z$, then
(27)$$\begin{array}{c} g_{1}\left(j\right)=\sum_{j^{\prime }=0}^{\eta_{1}-1}{\hat{f}}_{i}\left(j^{\prime }\right). \end{array}$$$0\leq \lambda_{1}<\eta_{1}$: In general, we have
(28)$$\begin{array}{c} g_{1}\left(j\right)=\left(1-\left(\lambda_{1}-\left\lfloor \lambda_{1}\right\rfloor \right)\right)\cdot {\hat{f}}_{i}\left(\left\lfloor \lambda_{1}\right\rfloor \right)+\left(\eta_{1}-\left\lfloor \eta_{1}\right\rfloor \right)\cdot {\hat{f}}_{i}\left(\left\lfloor \eta_{1}\right\rfloor \right)+\sum_{j^{\prime }=\left\lfloor \lambda_{1}\right\rfloor +1}^{\lfloor \eta_{1}\rfloor -1}{\hat{f}}_{i}\left(j^{\prime }\right). \end{array}$$Let us consider three special cases:—If $\lambda_{1}\in Z$ and $\eta_{1}\notin Z$, then
(29)$$\begin{array}{c} g_{1}\left(j\right)=\left(\eta_{1}-\left\lfloor \eta_{1}\right\rfloor \right)\cdot {\hat{f}}_{i}\left(\left\lfloor \eta_{1}\right\rfloor \right)+\sum_{j^{\prime }=\lambda_{1}}^{\lfloor \eta_{1}\rfloor -1}{\hat{f}}_{i}\left(j^{\prime }\right). \end{array}$$—If $\lambda_{1}\notin Z$ and $\eta_{1}\in Z$, then
(30)$$\begin{array}{c} g_{1}\left(j\right)=\left(\left\lceil \lambda_{1}\right\rceil -\lambda_{1}\right)\cdot {\hat{f}}_{i}\left(\left\lfloor \lambda_{1}\right\rfloor \right)+\sum_{j^{\prime }=\left\lceil \lambda_{1}\right\rceil }^{\eta_{1}-1}{\hat{f}}_{i}\left(j^{\prime }\right). \end{array}$$—If $\lambda_{1}\in Z$ and $\eta_{1}\in Z$, then
(31)$$\begin{array}{c} g_{1}\left(j\right)=\sum_{j^{\prime }=\lambda_{1}}^{\eta_{1}-1}{\hat{f}}_{i}\left(j^{\prime }\right). \end{array}$$

### 4.3. Discussion

From the above analysis, it can be found that all *N* points of ${\hat{f}}_{i}\left(j\right)$’s are made use of to generate ${\hat{f}}_{i-1}\left(j\right)$’s, and such treatment is fair for every ${\hat{f}}_{i}\left(j\right)$. For this reason, we will formally refer to this method as *fair numerical algorithm*. Given $\sum_{j=0}^{N-1}{\hat{f}}_{i}\left(j\right)=N$, it is easy to know $\sum_{j=0}^{N-1}g_{0}\left(j\right)=\sum_{j=0}^{N-1}g_{1}\left(j\right)=N$. Hence, (Equation 15) satisfies the normalization property by itself and no renormalization step is needed after (Equation 15).

Let us briefly discuss the convergence of numerical algorithms. Let f(u) denote the asymptotic form of $f_{n,0}\left(u\right)$ as n→∞. According to (Equation 5), it is obvious that
(32)$$\begin{array}{c} f\left(u\right)={\left(1-p\right)}^{1-r}f\left(u{\left(1-p\right)}^{-r}\right)+p^{1-r}f\left(\left(u-\left(1-p^{r}\right)\right)p^{-r}\right). \end{array}$$For the rounding/linear numerical algorithms, as analyzed above, partial parts of f(u) have been lost, so the discrete version of (Equation 32) will not exactly hold, while for the fair numerical algorithm, since all information of f(u) is reserved, the discrete version of (Equation 32) will exactly hold as the number of segments *N* goes to infinity.

## 5. Experimental Results

We use a classical CCS to compare three numerical algorithms. Given p=r=0.5, for *n* sufficiently large, as *i* increases, $f_{n-i}$ will tend to be ladder-shaped and we have [27,28].
(33)$$\begin{array}{c} \lim_{i\to \infty }f_{\infty -i}\left(u\right)=\left\{\begin{array}{cc} \frac{u}{3\sqrt{2}-4}, & 0\leq u<\sqrt{2}-1\\ \frac{1}{2-\sqrt{2}}, & \sqrt{2}-1\leq u<2-\sqrt{2}\\ \frac{1-u}{3\sqrt{2}-4}, & 2-\sqrt{2}\leq u<1 \end{array}\right.. \end{array}$$For simplicity, we set the final CCS $f_{n}\left(u\right)=\Pi \left(u\right)$, where Π(u) is a uniform function over [0,1). Some results are included in Figure 1, Figure 2 and Figure 3. In these figures, we set n=128 and N=1024. It can be observed that for small *i* (e.g., 1, 2, and 3), the rounding numerical algorithm performs almost as well as the fair numerical algorithm, while the linear numerical algorithm performs very poorly as there are some big spikes. When i=8, small spikes are also observed for the rounding numerical algorithm, but no spike is observed for the fair numerical algorithm. Finally, for large *i* (e.g., 16, 32, and 64), the resulting curves of the rounding numerical algorithm always fluctuate slightly along the real CCS given by (Equation 33), and more experiments show that there is no trend that these curves will finally converge to the real CCS as *i* increases. On the contrary, for both the linear numerical algorithm and the fair numerical algorithm, their curves coincide with the real CCS very well for large *i*, and there is a trend that these curves will finally converge to the real CCS as *i* increases.

## 6. Conclusions

As an important property of DAC, CCS finds its broad applications in many scenarios. However, CCS is usually incalculable so we have to resort to numerical algorithms. This paper finds that the original numerical algorithm proposed in our previous papers is not sound in theory and does not work well in practice (fails to generate an accurate approximation of CCS). Based on a strict theoretical analysis, this paper proposes a novel numerical algorithm that overcomes the weaknesses of the original numerical algorithm. The superiority of the newly-proposed numerical algorithm is well validated by experimental results.

A software package of source codes to reproduce the experimental results in this paper has been released in [33].

## Figures and Tables

**Figure 1 entropy-25-00437-f001:**
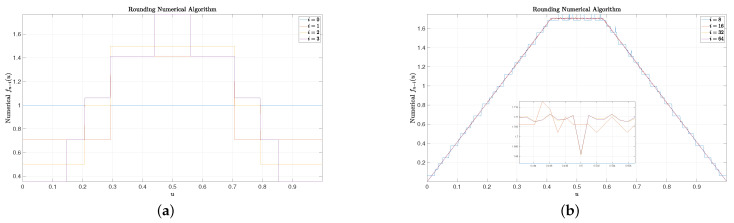
Some examples of the rounding numerical algorithm. Though the rounding numerical algorithm performs well for small *i*, it always fluctuates slightly along the real CCS for large *i* and will not finally converge to the real CCS as *i* increases. (**a**): Rounding numerical algorithm for ending symbols. (**b**): Rounding numerical algorithm for starting symbols.

**Figure 2 entropy-25-00437-f002:**
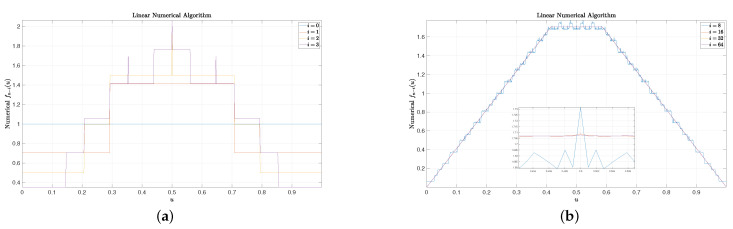
Some examples of the linear numerical algorithm. Though the linear numerical algorithm performs well for large *i*, i.e., the curves coincide with the real CCS well, it will cause big spikes for small *i*. (**a**): Linear numerical algorithm for ending symbols. (**b**): Linear numerical algorithm for starting symbols.

**Figure 3 entropy-25-00437-f003:**
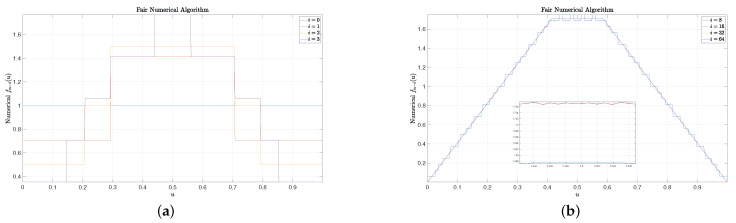
Some examples of the fair numerical algorithm. It overcomes all weaknesses of the rounding/linear numerical algorithms. (**a**): Fair numerical algorithm for ending symbols. (**b**): Fair numerical algorithm for starting symbols.

## Data Availability

Not applicable.

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
