# Peer review of "Fair Numerical Algorithm of Coset Cardinality Spectrum for Distributed Arithmetic Coding"

_entropy, 2023, doi:10.3390/e25030437_

Round 1

Reviewer 1 Report

Distributed source coding is a classic topic but still worth discussing. In this paper, a fair numerical algorithm of DAC, CCS is proposed with theoretical analysis and the numerical result is presented, together with the numerical result of rounding and linear algorithm. Therefore, the superiority of the fair algorithm is demonstrated. The theoretical part of this paper is verified to be correct. As a result, the achievement of this paper, i.e., the fair numerical algorithm, is thought to be meaningful.

Here are some suggestions:

l  Titles of reference [10] and [13] are capitalized for every first letter.

l  The superiority of DAC over traditional LDPC, Turbo and polar codes can be pointed in part 1.

l  For the last sentence in paragraph 2, part 1, a comma before “but” is missed.

Author Response

See the attached response file.

Reviewer 2 Report

This paper works on how to calculate the Coset Cardinality Spectrum (CCS) of Distributed Arithmetic Coding (DAC). A new algorithm is proposed, which can converge to the real CCS. This paper is well written in general, but some revisions are suggested. 

1 More ideas about CCS are suggested. It is not very clear why and how a number which shows how DAC coset cardinality is distributed can derive correct decoding formulae. For reviewers who are not so familiar with DAC and CCS, there is an obvious gap in understanding here. 

2 More ideas about the methodology of this paper are suggested. Intuitively, it is not clear why the proposed algorithm can converge to real CCS but previous ones cannot. 

3 More explanations about the problem setup are suggested. It is not clear why it is a backward induction, and why the final CCS is easier to obtain. 

4 Some pseudo code for the proposed algorithm is suggested. The reviewer believes that the pseudo code can make the algorithm easier to operate. 

5 More explanations on the figures are suggested. Specifically, the reviewer suggests that details of each figure should be illustrated respectively. Now, though the general purpose of the experiments is clear, but each individual figure is not clear enough. 

6 More literature citations are encouraged. Now only 14 papers are cited. The reviewer suggests the number to be doubled. 

All in all, this is a very interesting paper and worthy to be shared. The reviewer is happy to help with the next round review of this paper if there is. 

Author Response

Please see the attached response file.
